



# Mediterranean Sea heat uptake variability as a precursor to winter precipitation in the Levant

Ofer Cohen[1], Assaf Hochman[1], Ehud Strobach[2], Dorita Rostkier-Edelstein[1,3], Hezi Gildor[1], and Ori Adam[1]

[1]Fredy and Nadine Herrmann Institute of Earth Sciences, The Hebrew University of Jerusalem, Jerusalem, Israel
[2]Holon Institute of Technology, Holon, Israel
[3]Agricultural Research Organization, Volcani Institute, Rishon LeTsiyon, Israel

**Correspondence:** Ori Adam (ori.adam@mail.huji.ac.il)

**Abstract.** The Eastern Mediterranean is experiencing severe warming and drying, associated with global warming, making seasonal prediction of precipitation in the region imperative. Given that the Mediterranean Sea is the primary source of regional moisture and synoptic variability, here we explore the observed relation of Mediterranean Sea variability to Levant land precipitation during winter – the dominant wet season. Using Self-Organizing Map objective analysis, we identify three

dominant modes of sea surface temperature (SST) and ocean heat uptake variability in the Mediterranean Sea. Of these, two modes characterized by east-west variations are found to be statistically related to winter land precipitation in the Levant. Based on these relations, we define an Aegean Sea heat uptake anomaly index (AQA), which is strongly correlated with Levant winter precipitation. Specifically, AQA values during August are found to predict Levant precipitation in the following winter ($R = -0.60$). Wetter winters over the Levant following negative August AQA values are associated with more persis-

tent eastward-propagating Mediterranean storms, driven by enhanced baroclinicity and a stronger subtropical jet. The results present AQA as a useful seasonal predictor of Levant winter precipitation, and indicate that the representations of processes affecting Mediterranean cyclones, the subtropical jet, and ocean-atmosphere heat exchange, are key for seasonal forecasting skill in the Levant.

## 1 Introduction

The Eastern Mediterranean (EM) is generally recognized as a global warming "hotspot", projected to experience significant climatic changes, including rising temperatures and intensified droughts, as well as extreme precipitation events and flooding (Giorgi, 2006; Lionello et al., 2006; Lelieveld et al., 2012; Cramer et al., 2018; Zittis et al., 2022; Hochman et al., 2022a). Seasonal prediction of precipitation in the Levant, a region prone to water stress, is therefore crucial for adaptation efforts. Given that the Mediterranean Sea is a critical source of moisture and a key driver of synoptic variability influencing precipitation

in the Levant, we investigate the observed impact of spatiotemporal variability in the Mediterranean Sea on winter precipitation in the Levant.

    The EM lies in a transitional climate zone, subtended by temperate regions to the north and arid regions to the south (Goldreich, 2003; Ziv et al., 2006). EM climate is marked by relatively dry summers and wet winters, with the majority



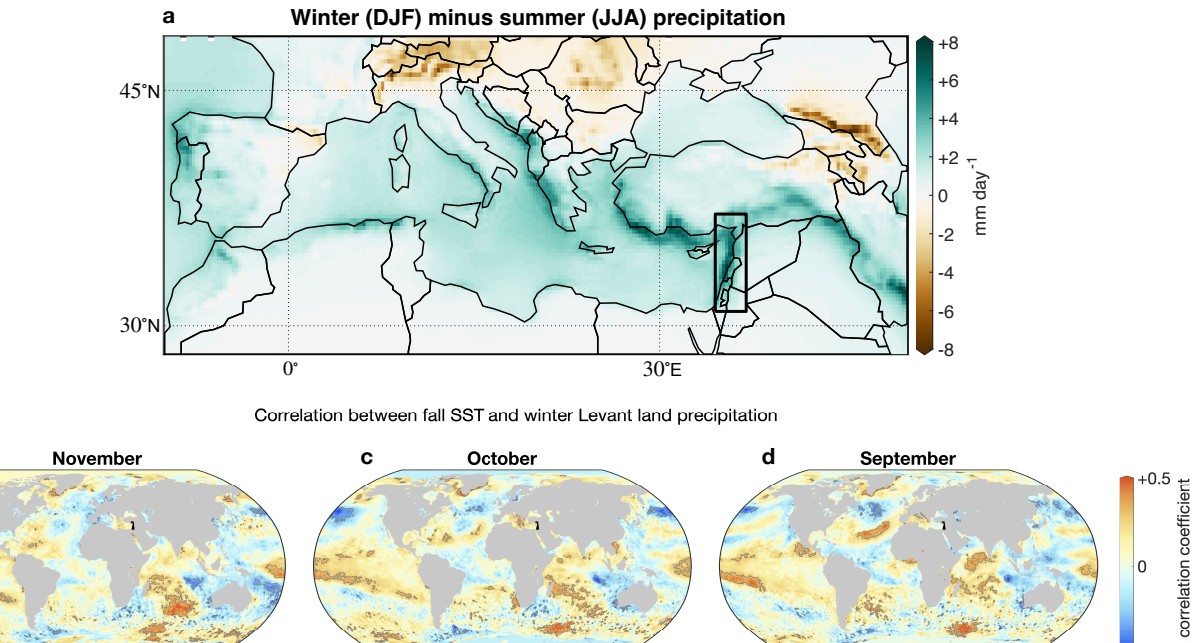

**Figure 1.** (**a**) Winter (December–February) minus summer (June–August) precipitation in the Mediterranean region; a black rectangle demarcates the Levant region considered in this study. (**b–d**) Pearson correlation coefficient of winter land precipitation in the Levant region and global sea surface temperature (SST) in the preceding November (**b**), October (**c**), and September (**d**) months, for the period 1979–2023. Data taken from the ERA5 reanalysis (Hersbach et al., 2020, see Section 2.1). Correlation 95% confidence bounds are shown in gray contours.

of precipitation occurring from December to February (Figure 1a). Seasonal synoptic patterns in the region result from the

interaction of large-scale systems, such as the subtropical jet, subtropical highs, and the Asian monsoon, with smaller regional systems, and are modulated by the conditions in the Mediterranean Sea (Eshel and Farrell, 2000; Goldreich, 2003; Alpert et al., 2004b; Ziv et al., 2006; Lionello et al., 2006; Saaroni et al., 2010; Hochman et al., 2022b). The interaction of global and regional systems, therefore, critically affects precipitation predictability in the region (Baruch et al., 2006).

     Various indices based on surface and atmospheric conditions in the Mediterranean have been used to capture precipitation

and temperature variations in the Mediterranean basin (Conte et al., 1989; Palutikof, 2003; Martin-Vide and Lopez-Bustins, 2006; Criado-Aldeanueva and Soto-Navarro, 2013; Redolat et al., 2019). In particular, multiple versions of a Mediterranean Oscillation index have been examined, motivated by the characteristic atmospheric east-west dipole in the Mediterranean basin (Conte et al., 1989). These, however, have shown limited predictive value in the EM on seasonal timescales (Redolat et al.,





2019). In contrast, statistical approaches incorporating delayed interactions and ocean-atmosphere heat fluxes have shown

significant potential for improving seasonal forecasting in the EM (Redolat and Monjo, 2024).

Recent work has demonstrated a delayed response of Levant precipitation to large-scale variations, which can serve as a basis for improved predictions of seasonal precipitation in the region (Amitai and Gildor, 2017; Hochman et al., 2022b, 2024). For example, statistical relations were established between Mediterranean Sea heat content in autumn and subsequent winter precipitation in several cities across Israel (Tzvetkov and Assaf, 1982; Amitai and Gildor, 2017). Synoptic weather systems in

the EM were also shown to be modulated by the Mediterranean Sea, with Mediterranean SST changes affecting the development and intensity of Mediterranean cyclones, thereby delaying inland winter precipitation peaks from early to late January compared to coastal regions (Ziv et al., 2006; Flaounas et al., 2018). Such variations in sea surface conditions are expected to have both dynamic and thermodynamic impacts on EM precipitation, influencing both atmospheric flow and regional synoptic conditions, as well as the thermodynamic properties of advected air parcels (Seager et al., 2014; Elbaum et al., 2022;

Tootoonchi et al., 2024; Seager et al., 2024).

Remote sea surface and atmospheric conditions are known to impact Levant winter precipitation. For example, Figure 1b-d shows global lagged correlation maps of observed SST and winter Levant precipitation. Significant lagged correlations are seen in various regions, including the tropical Pacific, the North Atlantic, and the Indian Ocean. The positive correlation in the North Atlantic during September (Figure 1d) agrees with previous works showing links between positive phases of the

North Atlantic Oscillation (NAO) and Arctic Oscillation (AO) and winter precipitation in the Levant, mediated by the effect of NAO and AO on the intensity of winter storms in the EM (Eshel and Farrell, 2000; Black, 2012; Givati and Rosenfeld, 2013; Luo et al., 2015), as well as upstream amplification extratropical cyclones originating in the North Atlantic (Raveh-Rubin and Flaounas, 2017). The lagged correlations in the Tropical Pacific (Figure 1b-d) are consistent with previous work linking ENSO and winter precipitation in northern Israel during the second half of the 20th century (Price et al., 1998). Similarly, SST

variability in the Indian and Pacific Oceans has been shown to be dynamically linked to sub-seasonal precipitation variability in the Levant (Hochman et al., 2024; Hochman and Gildor, 2025).

Significant lagged correlations are seen in the Mediterranean Sea, which are the focus of the present analysis. In particular, the lagged spatial correlation patterns in the Mediterranean Sea vary across months, suggesting non-trivial regional links between Levant precipitation and Mediterranean Sea variability, which are explored here. Specifically, we aim to: (i) explore the

observed links between objectively determined patterns of variability in the Mediterranean Sea and Levant winter precipitation; and (ii) analyze the physical processes underlying these links. Our results point to key features in the Mediterranean Sea that precede Levant winter precipitation, potentially providing a basis for improved seasonal prediction.

## 2   Data and methods

Our methodology is based on calculating the observed dominant spatiotemporal patterns of surface heat balance variability in

the Mediterranean Sea using objective methods, and identifying which elements of these modes of variability hold predictive power for Levant precipitation. This is then followed by an analysis of the regional moisture balance (Seager et al., 2010, 2014)



and synoptic conditions (Alpert et al., 2004b), providing context for the lagged response of Levant precipitation to Mediterranean Sea variability. The data, ocean mixed-layer heat balance, objective analysis methods, and analyses of moisture balance and synoptic conditions are briefly described below.

## 2.1 Data

For atmospheric and surface data, we use monthly and daily data from the European Center for Medium-Range Weather Forecasts (ECMWF) ERA5 atmospheric reanalysis at $0.25° \times 0.25°$ grid-spacing, covering the period 1979–2023 (i.e., post-satellite era; Hersbach et al., 2020). ERA5 data has been shown in previous studies to provide reliable estimates of *in situ* observations of the hydrological cycle (Seager et al., 2024; Tootoonchi et al., 2024). However, we validate our results using *in situ* data from rain gauges distributed throughout Israel provided by the Israel Meteorological Service (IMS, *https://ims.gov.il*). In accordance with ERA5, the reference observed sea surface temperature (SST) data are taken from the HadISST2 dataset (Titchner and Rayner, 2014) up to September 2007 and from the Operational SST and Ice Analysis (OSTIA) dataset (Good, 2022) thereafter. For precipitation over land, we use ERA5-land (Muñoz-Sabater et al., 2021) at approximately 9 km resolution, which provides an improved representation of land-specific processes.

## 2.2 Ocean mixed-layer energy balance

Upper-ocean heat content, given by the product of sea-water heat capacity $c_p$, ocean mixed-layer depth ($h_{ml}$, MLD), and SST ($T$), has been shown in previous work to be a contributing factor to processes affecting Levant precipitation over land, such as ocean-atmosphere heat and moisture exchange and land-ocean temperature contrasts (Amitai and Gildor, 2017; Tzvetkov and Assaf, 1982). In particular, Tzvetkov and Assaf (1982) and Amitai and Gildor (2017) demonstrated the potential of utilizing the ocean's upper layer heat content during fall months to enhance predictions of winter precipitation in the Levant. Building on these results, we hypothesize that changes in Mediterranean SST and ocean heat uptake are linked to precipitation changes in the Levant. However, since estimates of MLD can vary significantly across datasets and methodologies (Kara et al., 2000; d'Ortenzio et al., 2005; Treguier et al., 2023; Keller Jr et al., 2024), we avoid using MLD as a predictor.

Specifically, the energy balance equation of the ocean mixed layer can be written as (Gill, 1982)

$$c_p h_{ml} \dot{T} + c_p \nabla \cdot (\tilde{\mathbf{u}} T) = Q_f + K_z \tag{1}$$

where $\dot{T}$ is SST tendency, $\tilde{\mathbf{u}}$ is the vertical-mean horizontal flow in the mixed layer, $Q_f$ is the net downward heat flux at the ocean surface, and $K_z$ is the input of heat to the mixed layer from below, associated primarily with small-scale vertical mixing.

The net surface energy input into the ocean mixed layer ($Q_f$) consists of the net downward shortwave ($SW$) and longwave ($LW$) surface radiation, and the downward sensible ($SH$) and latent ($LH$) surface heat fluxes, minus the fraction of the shortwave radiation penetrating below the mixed layer ($Q_P$)

$$Q_f = SW + LW + SH + LH - Q_P \tag{2}$$



where $Q_P$ is calculated as a 25-meter $e$ folding decay of shortwave radiation, given by the equation $Q_P = 0.45SW e^{-\gamma h_{ml}}$, with $\gamma$ being a decay rate constant equal to $0.04 m^{-1}$ (Wang and McPhaden, 1999; Vallès-Casanova et al., 2025).

## 2.3 Self-Organizing Map (SOM) Analysis

To capture the spatial patterns of Mediterranean Sea variability, we employ the Self-Organizing Map (SOM) unsupervised neural network technique for clustering and visualizing high-dimensional data (Kohonen, 1990). For our particular analysis, SOM is advantageous over more traditional methods such as empirical orthogonal function (EOF) analysis because it does not require the patterns extracted to be orthogonal, potentially parsing the relevant phase space with fewer patterns than EOF. Nevertheless, qualitatively similar results, albeit statistically weaker, were obtained by deriving the patterns of Mediterranean
variability using EOF analysis.

The SOM algorithm is applied to detrended monthly deviations from the climatological seasonal cycle. The SOM parameters are optimized to maximize the correlation between the derived spatial patterns and Levant land precipitation, while minimizing the overall number of patterns. This yields an analysis based on three SOM patterns, applied to both SST and $Q_f$ (see Supplementary Materials for more details on the SOM parameters and procedures). The usage of three EOF patterns yields similar
optimal results, indicating that the emergent key patterns are not sensitive to the specific SOM algorithm parameters. Our use of monthly data for the SOM analysis increases the sensitivity to slowly evolving modes of variability in the Mediterranean (i.e., monthly time scales). Nevertheless, repeating our SOM analysis using 5-day ocean data yielded nearly identical spatiotemporal patterns, supporting the robustness of the derived patterns of variability.

## 2.4 Decomposition of precipitation variations

The steady moisture equation of the atmosphere can be written as (Seager et al., 2010)

$$\bar{P} = \bar{E} - \langle \nabla \cdot (\bar{\mathbf{u}}\bar{q}) \rangle - \langle \nabla \cdot (\overline{\mathbf{u}'q'}) \rangle - \overline{q_s \mathbf{u}_s} \cdot \nabla \bar{p}_s \tag{3}$$

where $P$ and $E$ represent precipitation and evaporation, respectively, $\mathbf{u}$ is the wind vector, $q$ denotes specific humidity, $p$ denotes pressure, and the subscript $(\cdot)_s$ denotes surface values. Angled brackets denote mass-weighted vertical integrals,

$$\langle \cdot \rangle \equiv \frac{1}{\rho_w g} \int_0^{p_s} (\cdot)\, \mathrm{d}p$$

where $\rho_w$ is the density of water and $g$ is Earth's gravitational acceleration; overbars and primes denote monthly temporal means and deviations thereof, respectively.

Following the methodology and terminology described in Seager et al. (2010), it follows from Eq. (3) that changes in precipitation can be decomposed into those involving changes in evaporation, the mean wind field (dynamic changes), the mean moisture field (thermodynamic changes), surface moisture transport, and transient eddies (Seager et al., 2010, 2014;
Kaspi and Schneider, 2013; Wills et al., 2016; Elbaum et al., 2022; Tootoonchi et al., 2024).



We define $\delta$ as the difference between two composites of monthly means,

$$\delta \equiv \overline{\overline{(\cdot)}}_2 - \overline{\overline{(\cdot)}}_1 \tag{4}$$

where a double overbar denotes a temporal average over some period. Neglecting changes associated with surface pressure, we rewrite the moisture balance equation to express the difference between two composites of winters,

$$\delta P \cong \delta E - \delta\langle \nabla \cdot (\bar{\mathbf{u}}\bar{q})\rangle - \delta\langle \nabla \cdot (\overline{\mathbf{u}'q'})\rangle. \tag{5}$$

In the next section, we refer to the second and third right-hand side terms as the changes in the mean and transient components of precipitation, respectively, where the mean component is further decomposed into mean thermodynamic ($-\langle \nabla \cdot (\bar{\mathbf{u}}[\delta\bar{q}])\rangle$) and the mean dynamic ($-\langle \nabla \cdot ([\delta\bar{\mathbf{u}}]\bar{q})\rangle$) components.

## 2.5    Semi-Objective Synoptic Classification

Due to the critical influence of synoptic-scale conditions on precipitation in the Levant (Lionello et al., 2006; Goldreich, 2003), we examine the relationship between these conditions and our calculated patterns of variability. Specifically, we classify synoptic conditions based on the semi-objective methodology proposed by Alpert et al. (2004b), which has been used in previous works to study EM seasonal synoptic variations (Alpert et al., 2004a), as well as changes under global warming in weather patterns (Hochman et al., 2018b, a, 2020) and extreme weather events (Rostkier-Edelstein et al., 2016; Hochman et al.,
2022a).

The classification method uses four single-level atmospheric fields: geopotential height, temperature, and the zonal and meridional components of wind; all at 1000 hPa and at a grid-spacing of $2.5°\times2.5°$ over the EM region (defined within the coordinates $27.5°$–$37.5°$N and $30°$–$40°$E), sampled at 12:00 UTC each day. The synoptic classification of each day within the EM is therefore calculated using 100 data points ($5 \times 5$ grid points for each of the four atmospheric fields), which are also
standardized by subtracting the long-term mean and dividing by the standard deviation of the time series of each field. The classification of each day to its synoptic type is performed by comparing each of these daily datasets to 426 days that were manually classified by a team of expert meteorologists (335 days from 1985 and 91 days from the winter of 1991–1992). The synoptic type of each day is determined as that for which the Euclidean distance from each of the manually classified days is lowest. The reference 426 days are categorized into five synoptic groups (Alpert et al., 2004b), of which two are associated
with winter precipitation:

     i   **Cyprus Low (CL):** A Mediterranean cyclone centered near Cyprus, often responsible for significant rainfall and stormy weather in the Levant region.

     ii   **Red Sea Trough (RST):** A low-pressure trough extending from the southern Red Sea into the EM, often associated with warm, moist air advection and convective activity. It produces occasional intense precipitation during transition seasons
in the southeastern Levant.

The remaining three synoptic groups are associated with warm and dry conditions:





iii. **Highs (H)**: A high-pressure system over the EM throughout the year (as an extension of the Siberian high), leading to stable, dry, and clear weather.

iv. **Persian Through (PT)**: A thermal low originating over the Persian Gulf and extending westward, typically bringing warm and moist conditions to the EM during summer.

v. **Sharav Low (SL)**: A transient heat low forming over the western Sahara and moving eastward, bringing hot, dry, and windy conditions to the EM, typically during spring.

To better identify the driving factors impacting the synoptic change, we analyze the regional atmospheric conditions between winter composites. Accordingly, we study the changes in mean sea-level pressure conditions, the 500 hPa geopotential, and the strength and position of the subtropical jet. Additionally, to quantify the difference in the regional baroclinic conditions, we use the Eady Growth Rate, as defined in Hoskins and Valdes (1990):

$$\sigma = 0.31 f N^{-1} |\partial_z \mathbf{v}| \tag{6}$$

Where $\sigma$ is the Eady Growth Rate between the 850 and 500 hPa pressure levels, $f$ is the Coriolis parameter, $N$ is the Brunt-Väisälä frequency, and $\partial_z \mathbf{v}$ is the vertical shear of the horizontal wind.

## 3 Results

We begin by calculating the spatiotemporal patterns of variability in Mediterranean SST and surface heat uptake ($Q_f$). Based on these patterns, we define an index that captures a strong statistical relation between summer $Q_f$ anomalies in the Aegean Sea and winter precipitation in the Levant. We then provide context for this relation using analyses of the regional hydrological cycle and synoptic conditions.

### 3.1 Spatiotemporal patterns of variability

We cluster the Mediterranean SST and $Q_f$ monthly time series into three SOM patterns of variability (Figures 2a and 3a, respectively). The SOM patterns for SST and $Q_f$ show a high degree of similarity, allowing their joint analysis. SOM pattern 1 explains 14% of SST temporal variance and 13% of $Q_f$ temporal variance, and can be described as generally capturing a gradient between the central Mediterranean and its eastern and western parts. This pattern does not show a significant lagged correlation to Levant precipitation (Figures 2b and 3b), and is therefore not further analyzed.

The second and third SOM patterns of SST and $Q_f$, which account for most of the temporal variance (45% and 42% for SST and 46% and 35% for $Q_f$, respectively), generally capture east-west gradients across the Mediterranean basin (Figures 2a and 3a), consistent with the characteristic east-west dipole seen in atmospheric variables (Conte et al., 1989). Specifically, Pattern 2 features a node located between the Ionian and Tyrrhenian Seas (i.e., east and west of Sicily); Pattern 3 exhibits a pronounced gradient between the western Mediterranean and the Aegean Sea. For both SST and $Q_f$, Patterns 2 and 3 have significant lagged correlations with Levant winter precipitation (peaking during November, September, and July for SST, and





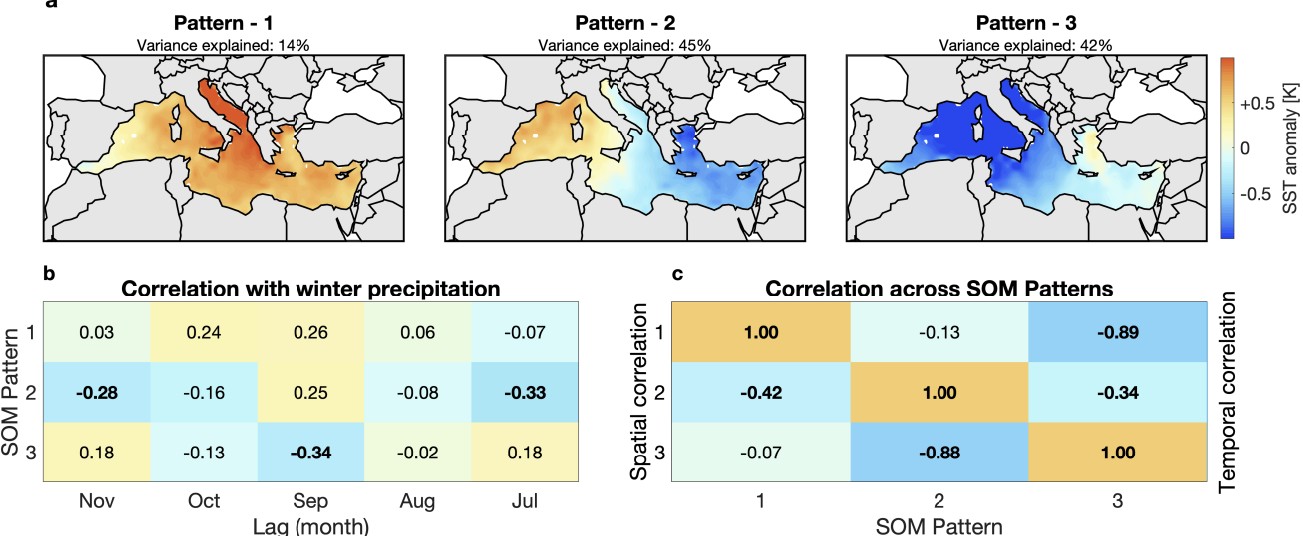

**Figure 2.** (a) Mediterranean SST monthly time series clustered into three SOM patterns; (b) Lagged correlation of each monthly SOM pattern with Levant winter precipitation over land; (c) Spatial (below diagonal) and temporal (above diagonal) correlations across the SOM patterns. Pearson correlation coefficients significant at the 5% level are in bold. Data taken from ERA5 for the period 1979–2023. The variance explained by each SOM pattern refers to the temporal variance explained by each pattern.

during October and August for $Q_f$), showing potential for sub-seasonal to seasonal prediction (Figures 2b and 3b). Consistent with Eq. (1), since $Q_f$ drives variations in SST, the peak correlation of $Q_f$ is observed to lag that of SST by an additional month. Moreover, despite this extended lag, the highest correlation with Levant precipitation is found for Pattern 2 of $Q_f$ in August ($R = 0.53$).

The spatial and temporal correlations across the SOM patterns are shown in Figures 2c and 3c. For both fields, Patterns 2 and 3 are strongly spatially correlated ($|R| \geq 0.78$) but relatively weakly temporally correlated ($|R| \leq 0.36$), indicating that despite their topographical similarities, these patterns vary on different timescales.

Given that the largest correlation with Levant winter precipitation is found for the $Q_f$ SOM patterns, we hereon focus our analysis on $Q_f$. During the peak correlation month of August, $Q_f$ variations are dominated by latent heat fluxes, with minor contributions from sensible heat fluxes and negligible contributions from radiative fluxes (Supporting Materials Figure S1). Since latent and sensible heat fluxes are strongly dependent on near-surface winds and atmospheric conditions (such as temperature and humidity), this suggests that coupled ocean-atmosphere processes are critically linked to the lagged correlation.



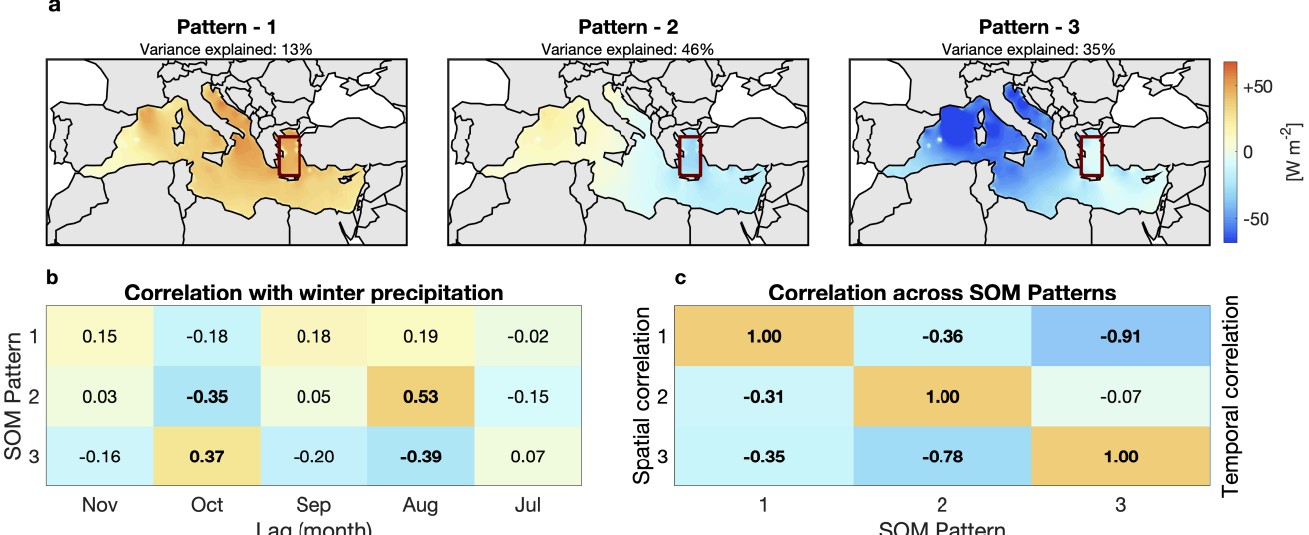

**Figure 3.** (a) Mediterranean $Q_f$ monthly time series clustered into three SOM patterns. Red rectangle indicates region used to calculate the Aegean $Q_f$ Anomaly index (AQA, defined in section 3.2); (b) Lagged correlation of each monthly SOM pattern to Levant winter land precipitation; (c) Spatial (bellow diagonal) and temporal (above diagonal) correlations across the SOM patterns. Pearson correlation coefficients significant at the 5% level are in bold. Data taken from ERA5 for the period 1979–2023. The variance explained by each SOM pattern refers to the temporal variance explained by each pattern.

## 3.2 Aegean $Q_f$ Anomaly index

We find that mean $Q_f$ values in the Aegean Sea reproduce the lagged correlations with Levant precipitation seen for SOM patterns 2 and 3. We therefore define an Aegean Sea $Q_f$ Anomaly index (AQA) as a precursor to Levant winter land precipitation. Specifically, AQA is defined as the $Q_f$ detrended anomaly from seasonal climatology in the north-eastern region demarcated in Figures 3a and 4(a-b) (23.5°–26.5°E, 32.5°–35°N), normalized by the standard deviation of the anomaly timeseries. Note, however, that based on SOM patterns 2 and 3, this index does not only indicate $Q_f$ changes in the Aegean Sea, but also contrasting $Q_f$ changes in the western Mediterranean (and similarly for SST; Figures 2 and 3).

Monthly AQA values are shown in Figure 4c. AQA is unit-less, with positive values indicating higher ocean heat uptake in the Aegean Sea relative to climatological conditions (Figure 4a, b). AQA is strongly correlated with $Q_f$ Pattern 2 ($R = -0.85$, Figure 5a) and can therefore be interpreted as generally proportional to the amplitude of this SOM pattern.

AQA is significantly correlated with Levant winter precipitation, in both ERA5 data and *in situ* IMS rain gauges, with the strongest lagged correlation in August ($R = -0.60$), in agreement with the $Q_f$ Patterns 2 and 3 (Figure 5a). The correlation is strongest for winter (Dec–Feb), but is significant for each of the winter months (Figure 5b). These results are not sensitive



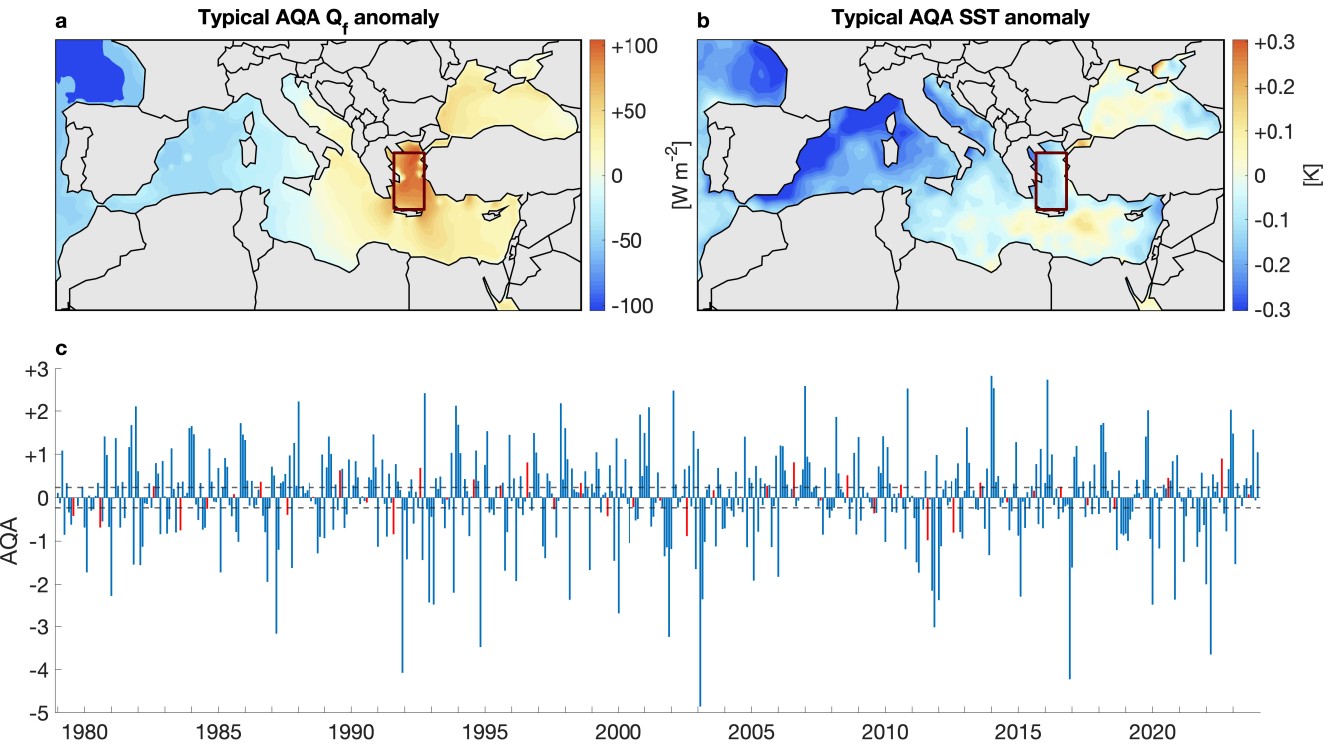

**Figure 4.** Aegean $Q_f$ anomaly Index (AQA), defined as the detrended anomaly from climatology of $Q_f$ in the Aegean Sea (23.5°–26.5 °E, 32.5°–35°N, red rectangles in panels a and b), normalized by the standard deviation of the anomaly timeseries. The typical $Q_f$ (a) and SST (b) difference between positive and negative AQA months (taken as months above or below ±0.5 the standard deviation of AQA). (c) Monthly AQA values from January 1979 to December 2023. August months, which are the most strongly correlated with winter Levant land precipitation, are shown in red. Gray dashed horizontal lines denote ±0.5 the standard deviation of August AQA values.




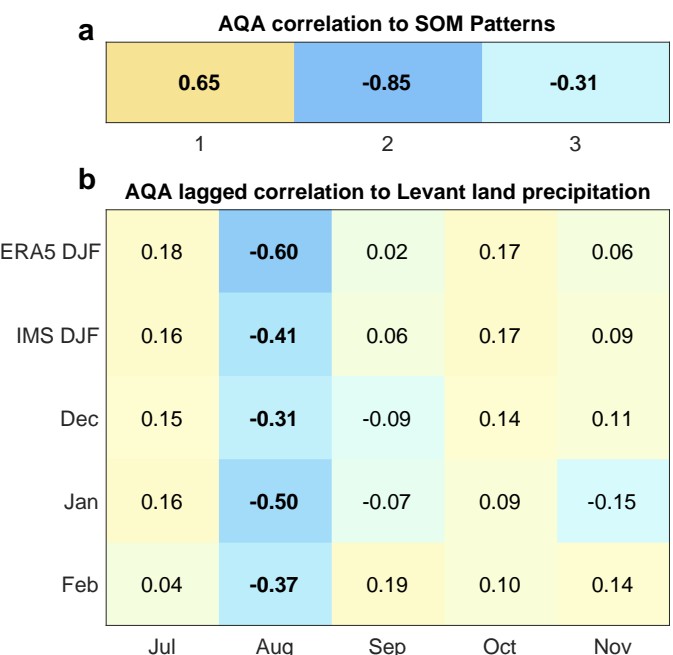

**Figure 5.** (a) Correlations of the Aegean $Q_f$ anomaly index (AQA) with the $Q_f$ SOM patterns. (b) Correlations of AQA with Levant land winter precipitation using ERA5-land data and rain gauge data from the Israel Meteorological Service (IMS, https://ims.gov.il).

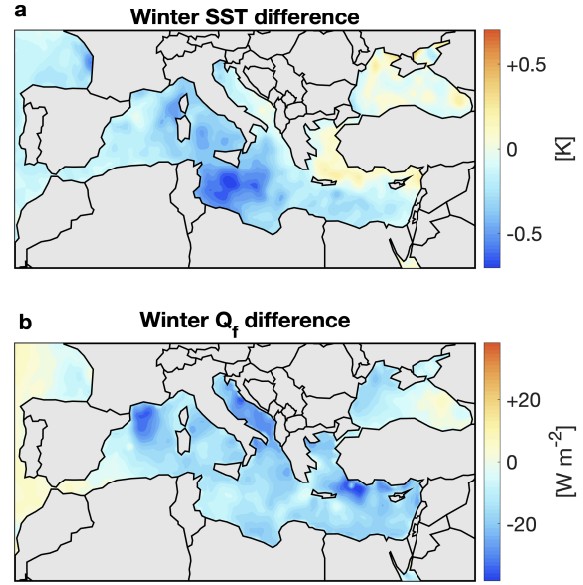

**Figure 6.** Mean (a) SST and (b) $Q_f$ difference between composites of winters preceded by August AQA values below and above half the standard deviation of August AQA values, respectively.





to the choice of Dec–Feb as the winter months, and remain significant for earlier, later, or longer winter months combinations (specifically, Nov–Mar, Nov–Jan, and Jan–Mar, shown in Supporting Materials Figure S2).

We now turn to analyzing the winter conditions associated with AQA variations by examining the differences between com-
posites of winters preceded by August AQA values below and above plus and minus half of the standard deviation of August AQA values (∼13 years in each composite group; cf. Figure 4). The difference in winter SST between the two composites, shown in Figure 6a, exhibits higher SST conditions in the north-eastern Mediterranean, which persist throughout winter (not shown). The increased winter SST in the north-eastern Mediterranean is indicative of increased upward surface heat and mois-ture fluxes, which, in turn, imply favorable conditions for cyclogenesis and storm intensification (e.g., Flaounas et al., 2022).
Accordingly, the winter $Q_f$ difference between these winter composites (Figure 6b) shows decreased ocean heat uptake in the EM, driven primarily by increased upward latent and sensible heat fluxes during negative AQA composite winters.

The composite difference of hydrological changes is shown in Figure 7. A significant increase in precipitation is seen in the EM (Figure 7a). The majority of the precipitation increase is seen in the mean component of the moisture flux convergence (Figure 7c), with minor contribution from transient eddies in the northern Levant (Figure 7d), and negligible contribution
from changes in evaporation (Figure 7b). Further decomposition of the mean component (Figure 7c) shows that the mean thermodynamic component (Figure 7e) is negligible, while the mean dynamic component dominates the precipitation response (Figure 7f).

We therefore conclude that the increased winter Levant precipitation associated with negative AQA anomalies during the preceding August is mediated by changes in regional mean flow patterns, creating more favorable conditions for precipitation
by synoptic systems migrating eastward from the central Mediterranean to the Levant. Next, we turn to synoptic analysis to diagnose the meteorological conditions underlying the winter precipitation response to AQA.

## 3.3 Synoptic analysis

Using the semi-objective synoptic classification (Alpert et al., 2004b), we assess the difference in synoptic conditions between winters preceded by August AQA above and below half the standard deviation of August AQA values (Figure 8a). Negative
August AQA values are associated with increased prevalence of Cyprus Lows (CL, 32 vs. 25 days per winter) and decreased prevalence of Red Sea Trough systems (RST, 27 vs. 35 days per winter), with negligible changes in the other synoptic groups. Given that winter precipitation in the EM is dominated by eastward propagating Mediterranean cyclones (i.e., Cyprus Lows; Saaroni et al., 2010), and that Red Sea Through synoptic conditions rarely lead to precipitation, these results are consistent with the winter precipitation response to AQA shown in Figure 7a.
To assess whether the enhanced Cyprus Low activity results from increased number or duration of storms, we calculate the composite difference in the number of synoptic systems occurring in the region during winter (defined as the number of days minus the number of consecutive days of each synoptic type during winter; Figure 8b). The number of Cyprus Low systems, as well as of Red Sea Trough and High systems, shows no significant sensitivity to AQA. This, in turn, indicates that the wetter Levant winters in response to negative AQA anomalies in the preceding August result from more persistent
precipitating Cyprus Low systems during winter.



**Figure 7.** The decomposed hydrological balance in the Mediterranean region. Bars denote monthly means, and primes denote transient variations. Dotted regions are 95% statistically significant using a bootstrapping test. (a) Changes in precipitation between negative and positive AQA index winter composites, respectively; (b) Changes in evaporation between composites; (c) Changes in the mean vertically integrated moisture balance; (d) Changes in the transient-eddy component of the moisture flux; (e) Changes in the mean thermodynamic component; (f) Changes in the mean dynamic component.





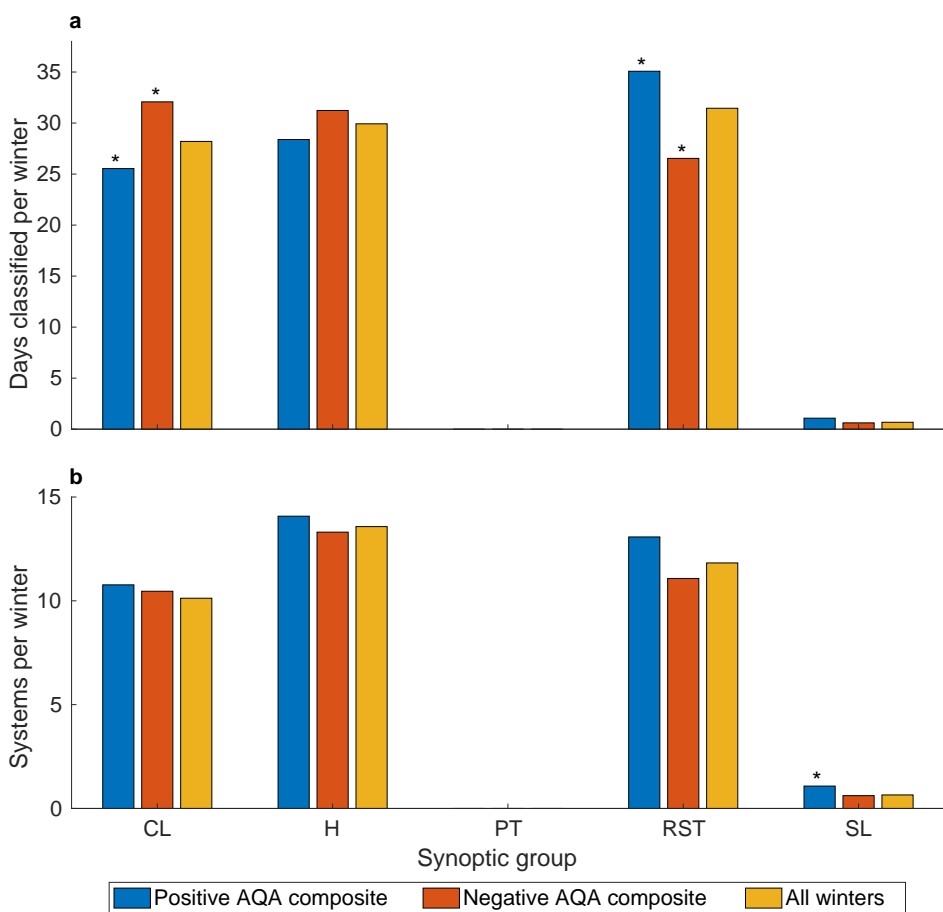

**Figure 8.** The difference in the number of winter days classified as each synoptic group (a), and the number of winter synoptic systems occurring (b), between composites of winters preceded by August AQA values above and below, plus and minus half of the standard deviation of August AQA values. Asterisks denote that the change is significantly different above the 90% threshold using the binomial significance test.





Since the hydrological decomposition pointed to changes in the mean flow as the primary driver of wetter Levant winters (Figure 7f), we now asses the relation of AQA to the prevailing regional westerlies. As shown in Figure 9a, negative AQA values in August are associated with a stronger subtropical jet over the EM during winter, which goes along with a low sea-level pressure anomaly over the Aegean Sea (Figure S4) and a negative 500hPa geopotential anomaly north of the Aegean

Sea (Figure 9c). This large-scale jet intensification coincides with increased upper-level divergence along EM storm tracks (Supporting Material, Figure S5) and a positive anomaly in the Eady Growth Rate (Figure 9e), both indicative of more favorable conditions for baroclinic convective instability, consistent with the increased precipitation in the Levant (Figure 7a).

In summary, our findings reveal that a negative ocean heat uptake anomaly in the Aegean Sea during August is a precursor to enhanced winter precipitation in the Levant. This link is dynamically mediated by the increased persistence of precipitating

Cyprus Low systems traversing the region, driven by a strengthening of the subtropical jet over the EM and a concurrent intensification of regional baroclinicity.

## 4   Summary and Discussion

The relation of Mediterranean Sea variability and winter precipitation in the Levant is explored. Objective analysis reveals that changes in the Mediterranean Sea heat uptake act as a precursor to inter-annual variability in Levantine winter precipitation.

Based on this, we define an Aegean Sea heat uptake anomaly index (AQA), representing anomalous ocean heat uptake ($Q_f$) in the Aegean Sea during August. AQA shows a significant negative correlation with subsequent winter land precipitation in the Levant ($R = -0.60$). The associated increase in precipitation is driven by the more persistent eastward-migrating Mediterranean storms, which constitute the dominant source of winter rainfall in the region. This increase is linked to a strengthening of the regional subtropical jet, promoting enhanced baroclinicity and more favorable conditions for storm development and

maintenance.

Specifically, using Self Organizing Map (SOM) analysis, we identify three dominant spatiotemporal patterns of variability in Mediterranean SST and $Q_f$ (Figures 2 and 3). Of these, the patterns characterized by east-west gradients are found to predict variations in Levant winter precipitation. The statistical relations are significant for both SST and $Q_f$, and are qualitatively reproduced for *in situ* data in Israel. Similar patterns of variability are produced with Empirical Orthogonal Function analysis

(not shown), albeit with weaker statistical relations. The $Q_f$ anomalies, which are generally anti-correlated with SST anomalies (Figure 4), are primarily driven by changes in latent heat fluxes (Hochman et al., 2022b), highlighting the important role of ocean-atmosphere interactions in Mediterranean Sea variability, and in particular in the lagged response of Levant precipitation.

Composite analysis of winters preceded by negative and positive August AQA values reveals a response characterized by:

   i. Enhanced precipitation in the Eastern Mediterranean (EM), particularly in the Levant and southern Turkey (Figure 7);

ii. Elevated SST in the northern parts of the EM, a deep thermal low centered north of the Aegean Sea (Figures S4 and 9c), and reduced $Q_f$ throughout the EM (Figure 6);

   iii. More persistent eastward migrating EM Mediterranean cyclones, commonly termed Cyprus Lows (Figure 8a);





**Figure 9.** Composite difference between winters (Dec–Feb) preceded by negative and positive August AQA values in (a) 250 hPa wind during winter, (c) geopotential height at the 500 hPa pressure level, and (e) Eady Growth Rate. Right column panels show the respective winter climatology. Data taken from ERA5 for the years 1979–2023. Stippling indicates 95% confidence estimated using a bootstrap test.



    iv. Strengthened regional subtropical jet, which goes along with more baroclinic conditions in the EM, and enhanced upper-level divergence (Figures 9, and S5).

A decomposition of the winter precipitation response indicates that the wetter Levant winters following August AQA values are associated with changes in the mean flow (Figure 7), in support the strengthening of the regional jet and associated baroclinic instability as the culprit of the wetter winters. The strengthening of the subtropical jet is consistent with geostrophic enhancement by the thermal low north of the Aegean Sea. However, additional confounding factors such as interaction with the polar jet and eddy heat and momentum fluxes may also play a role (Flaounas et al., 2022). Given that remote regions are

known to affect Levant precipitation (Figure 1), the relation of AQA and Levant precipitation may be mediated by contributing factors from outside the Mediterranean basin. However, we find no appreciable statistical relations between AQA and indices known to to be related to Levant precipitation, such as the North Atlantic Oscillation index (NAO), the Southern Oscillation index (SOI), and the SST anomaly in the NINO 3.4 region in the Pacific (Figure S3; Price et al., 1998; Black, 2012; Givati and Rosenfeld, 2013; Luo et al., 2015).

AQA, therefore, emerges as a potentially useful index for improving the skill of seasonal precipitation forecasts in the Levant, accounting for approximately one-third of inter-annual variability. In addition, the mechanisms linking AQA and Levant precipitation suggest that the representation in regional models of processes affecting Mediterranean cyclones, ocean-atmosphere heat exchange, and the subtropical jet, is key for improving seasonal forecasts (Flaounas et al., 2022; Redolat and Monjo, 2024). We intend to isolate these processes and assess their influence on forecast skill in future work.

*Author contributions.* Data analysis and writing were done by OC. OA and AH contributed to the conceptual derivation of the methodology and results. All other co-authors provided editorial contributions.

*Competing interests.* All authors declare no competing interests.

*Acknowledgements.* The research was funded by Grant 4749 of the Israeli Ministry of Innovation, Science, and Technology. AH acknowledges support by the Israel Science Foundation (grant #978/23), the Nuclear Research Center of Israel, and the Planning and Budgeting

Committee of the Israeli Council for Higher Education under the 'Med World' Consortium.





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
