# Peer review of "Mediterranean Sea heat uptake variability as a precursor to winter precipitation in the Levant"

_EGUsphere, 2025_

## Referee Comment (RC3)

Mediterranean Sea heat uptake variability as a precursor to winter precipitation in the Levant – review

The researchers investigated the seasonal predictability of winter precipitation in the eastern Mediterranean (EM) with respect to the Mediterranean SST and heat uptake in the preceding summer/autumn. Based on both reanalysis data and observations, the authors usie a SOM algorithm to classify SST and heat uptake fields into three groups. of them two show correlation to the following winter EM precipitation. Spatial patterns of lagged correlations between SST and $Q_f$ and subsequent winter precipitation in the EM are explored and compared between the two relevant groups, detecting peaks in August at the Aegean sea. The authors then define an Aegean sea anomaly index (AQA) which, when taken for August months, can act as a precursor with negative correlations to winter precipitation in the EM. Composites of positive and negative AQA-preceded winters are then further analyzed, thoroughly investigating the impact of the AQA on synoptic systems and on the decomposed hydrological balance.

The researchers conclude by proposing a cross-seasonal link between Aegean SSTs and $Q_f$ in August to winter precipitation in the EM. Through the eastward migration of the subtropical jet following negative AQA anomalies, cyclonic activity is reinforced in the EM, allowing for more persistent precipitation.

The study pursues a relevant objective in a novel approach, using reliable datasets and a range of sound methods. The manuscript is comprehensive and sound, well written and nicely structured, albeit slightly dispersed and over-informative at times.

I recommend accepting the paper after some minor concerns are answered and revisions made.

General comments:

1. There is a general feeling that the MS is constructed step by step based on its own statistical results. I recommend restructuring the MS in light of the results and the decisions taken by them, in a way that wouldn't overwhelm the reader. E.g., if the SOM shows similar results for both SST and $Q_f$, and seeing as $Q_f$ proves to be a better predictor, why not avoid showing and discussing SST throughout? Clearly a lot of effort was put into this work, but not all must be shown to the reader.
2. The SOM algorithm is underutilized here. The 1X3 network inherently looks for zonal variability only, leading to a rough separation that could have been easily obtained with less complex algorithms such as EOF or k-means, simplifying the interpretation of the results. Also, a significant test / standard deviations for the classified patterns is required to show regions of higher and lower confidence.

3. I think conciseness should be sought after in the revised MS. Some information can be removed from the figures and following discussions, increasing the focus on the main points the authors wish to convey.

Minor comments:

Figure 1: why is it important to show the difference between summer and winter precipitation? This panel isn't addressed in the text and the point of it is unclear. Also, the global maps are hard to see. Is it essential to have three of them? Perhaps one is enough to provide the context of remote teleconnections?

L52: amplification of...

L179 and elsewhere: the authors refer to cluster frequency as "explained variance". I find this terminology more suitable for EOF analysis. For clarity and fluidity, consider using cluster frequency (or a similar term) throughout.

L184: what do you mean by "node"? please clarify

Figure 2: I find the "correlation across SOM parameters" puzzling and unnecessary to propagate the reader through MS. Similarly, showing multiple months with mostly weak correlations that fluctuate from month to month does not support the robustness of the analysis. I suggest keeping the only the leading SOM input - $Q_f$ and only the most successful lead-time (August). This will make the message easier to take in.

L200: This is a methodological leap, especially since it is unclear how robust these spatial patterns are in the Aegean. More statistical testing is required to establish the Aegean Sea as an anchor for the SOM separation. E.G., did the results differ significantly when using the Ionian/Ligurian seas instead?

L207: anti-correlated?

Figure 4: would it be possible to denote the corresponding winter precipitation anomalies?

L275: please state that these conclusions relate to the negative AQA phase (correct?)

L280-283: this sentence is unclear. Please rephrase

L283: what do you mean by "geostrophic enhancement"?

L293: I would consider rewriting this section with emphasis on the importance of this work and less speculations and general ideas that do not relate directly to this research.

---

## Author Comment (AC1)

**Mediterranean Sea heat uptake variability as a precursor to winter precipitation in the Levant**
Cohen et al.

**Response to Reviewers**

We thank the reviewers for their helpful and insightful comments, which improved the analysis and discussion in the manuscript. To address the reviewer's comments, we adapted the main text of the manuscript and added additional sections and figures to the Supplementary Materials (SM) with additional analyses. For convenience, the revised text with tracked changes is provided below.

Specifically, we added the following findings to the SM:

1. We expanded the SM section S1, elaborating on the SOM analysis and optimization, and added analyses and explanations of the method used to select the final SOM structure. Specifically, we analyzed the amount of variance explained by the different SOM structures, by an individual SOM pattern within each structure, and the Topographic errors that occur for each structure. Using these, we explain our selection of the final SOM structure, aiming to maximize the variance explained by the SOM structure and the leading pattern, and to minimize the amount of topographic errors, for the minimal amount of patterns, consistently yielding 3 very similar leading patterns in both the SOM and EOF analyses .

2. We added a section in the SM (S3) presenting the EOF analysis of ocean heat uptake and SST, and showing their correlations with Levant winter precipitation. The results are similar to those of the SOM analysis, showing that our selected patterns are not sensitive to our choice objective methodology. We decided to focus on the SOM analysis due to its slightly higher correlations (particularly with SST), as a prerequisite for defining AQA, although the EOF analysis can also be used to justify the regions chosen for the definition of AQA.

3. We added a section in the SM (S2) analyzing the relationship between Aegean Sea SST anomalies and Levant winter precipitation, showing that a similar analysis and predictive behavior can be achieved using SST, but that the results tend to be less significant. Given SST's lower predictive skill, we focused on ocean heat uptake in our analysis.

4. We added spatial maps showing the significance of the SOM patterns (added to SM section S1, Figure S4). We analyzed the statistical significance of the correlations between AQA and Levant winter precipitation (SM section S5, Figure S17).

5. To assess the robustness of the Aegean Sea's spatial definition in the calculation of AQA, we changed the area used to define the Aegean Sea by shifting it by ±0.25 degrees north and east, finding that the area defined as the Aegean Sea does not

significantly affect the correlations to Levant winter precipitation, which is the main objective of our analysis (added to SM section S5).

Additionally, following the Reviewers' comments, we tried to provide more comprehensive and clearer descriptions of our results and the analyses we performed. Specifically:

1. We elaborated on the optimization process used for the SOM analysis in the methods section, explaining the SOM method (Section 2.3), incorporating the selection method we used for the final SOM structure - minimizing the overall number of patterns, while maximizing the amount of variance explained by the SOM patterns (individually and in total) and minimizing the topographic error.
2. We added additional details on the calculation of correlations, both across SOM patterns and between SOM patterns and Levant land winter precipitation, to the captions of Figures 2 and 3, to make the analysis more straightforward.
3. We revised the manuscript to explicitly emphasize that additional analysis is needed to fully explain the physical mechanism underlying the lagged atmospheric response in the eastern Mediterranean. We addressed this question by analyzing atmospheric and surface data, but were unable to identify a relevant process at this time. We therefore hypothesize that the underlying processes driving the lag are oceanic in nature. Further analysis of the lagged response, therefore, is beyond the scope of the present analysis, and we hope to address it in future work.

Specific responses (black) to the Reviewers' comments (blue) are provided below.

**Reviewer 1**
The manuscript concludes that the Mediterranean heat uptake, especially in the Aegean basin, can be used as a precursor to predict winter land precipitation in the Levant (eastern Mediterranean). The topic is interesting and the manuscript has the potential to be a useful contribution. But I think a few points need to be addressed before its publication.

We thank the reviewer for the insightful comments.

Firstly, many technical details are missing, which is harmful for a good understanding of the manuscript. For example, it is really difficult to figure out what presented in Figures 2b, 2c, 3b, and 3c (tables showing correlation coefficients).

A more comprehensive description of the selection process for the SOM final structure has been added to the SM, and we also expanded our explanation about the selection process in the main text. Additionally, the captions of Figures 2 and 3 now include a more detailed description of the correlation calculations performed across and between the SOM patterns, as well as between the SOM patterns and Levant winter precipitation.

Secondly, the use of SOM is unusual and not justified in this work. Normally SOM is used with a 2-D grid of different nodes, but there are only three patterns in the manuscript.

We agree that there's some redundancy in using only three SOM patterns. This, however, is an emergent result of the selection process we used, which yielded three patterns. The selection process and rationale for using SOM patterns are now clarified in the SM and main text. Specifically, we examined both EOF and SOM patterns, aiming to maximize the variance explained by the SOM/EOF patterns with the fewest possible patterns. We found three key modes of variability in the Mediterranean, with the SOM patterns showing somewhat higher correlations with winter Levant precipitation, potentially because they are not constrained to be orthogonal to each other (as opposed to EOFs), allowing them to capture more of the variability. Since our goal was to find the modes with the highest predictive power for Levant precipitation, we base our results on the SOM patterns. However, the results are essentially not sensitive to the use of either SOM or EOF patterns.

Finally, the manuscript concludes that AQA (Aegean Sea heat uptake anomaly) is a good indicator for precipitation anomalies in the Levant. But, compared to SST which is a state variable, the heat flux is much more difficult to measure or to be deduced from observation. Its usefulness might be quite limited.

To address our unconventional decision to focus on ocean heat uptake for the definition of AQA, instead of SST (which is a state variable and much easier to measure), we added a section in the supplementary materials analyzing the correlations of monthly Aegean Sea SST anomalies, instead of ocean heat uptake, with Levant winter precipitation. The results are similar to those obtained with ocean heat uptake, but are less statistically significant. We therefore chose to focus on Aegean Sea ocean heat uptake (AQA) for the moisture balance and synoptic analysis, instead of SST. In addition, as we show in the text, ocean heat uptake is proportional to SST tendency, which explains the extended lag of about a month when considering ocean heat uptake relative to SST. This again points to ocean heat uptake as a more potent predictor. To address this point in the manuscript, we added an explanation in the results (Section 3.2) to clarify our decision to focus on ocean heat uptake rather than SST.

Furthermore, it is also a little disappointing that a clear physical mechanism is missing to link the AQA to the precipitation in the target area.

We agree that a clear mechanism accounting for the lagged response is sorely missing. We now acknowledge in the text that a complete physical mechanism is not presented, and that we still lack the explanation for the lagged response between the sea surface pattern and the synoptic conditions.
Having said that, given the seasonal timescale of the regional lagged response, we hypothesize that oceanic processes likely play a leading role, which we hope to explore further in future analysis using ocean reanalysis data and Lagrangian trajectory analysis. Initial promising

results, presented in Figure 6, show that winters following negative AQA August anomalies tend to be warm in the north-eastern Mediterranean, possibly contributing to the more unstable conditions we observe in the synoptic analysis. Nevertheless, additional analysis is needed to reach a definitive conclusion.

There are a few other points:

1. Figure 1 is used to motivate the present work exploring the role of the Mediterranean Sea in modulating precipitation in the Levant. But it is not very convincing. The signal is not remarkable in the Mediterranean, but much stronger in other basins of the global ocean.

We use Figure 1 primarily to provide context for our review of existing work relating Levant precipitation to other regions (e.g., the Indian Ocean) and to show that there is considerable variation in the statistical relations observed in the leading months (Sep–Oct), suggesting nuance in the lagged relations. Indeed, somewhat surprisingly, SST correlations in the Mediterranean Sea are significant but lower than in more remote regions. But as we later show, these become significantly stronger when ocean heat uptake is considered.

2. Line 106, "The SOM algorithm is applied to detrended monthly deviations from the climatological seasonal cycle". It is not clear how SOM is performed. Is it applied to anomalous SST, i.e., SSTa(m7:m11,y1979:y2023)?

Yes, we perform the SOM analysis on anomalies from the seasonal cycle for the entire time series, which is also detrended over the 45 years.

Is there any coherence or consistency among m7 (July) to m11 (Nov) for a same year?

To address the concern of autocorrelations in the sea surface conditions, we calculated the correlation of the detrended anomaly from the seasonal climatology of SST in July and November, and the correlations between them across the Mediterraenan:

**correlation between SST detrended anomalies in Jul and Nov**

[Figure]

[Figure]

The correlations indicate that in the Aegean Sea, where we focus our analysis, they are relatively weak (below R=0.25). Other regions in the Mediterranean exhibit stronger autocorrelation between summer and fall SST. Specifically, the western Mediterranean, and the area around Sardinia in particular, shows stronger correlations between summer and fall SST conditions.

3. Figs 2 and 3, Figure Caption, "SST monthly time series". There is confusion for the term "time series". More precisions are needed.

To address this, we changed the phrasing in the captions of Figures 2 and 3, so that the data we used for the SOM analysis will be more clearly stated, eliminating the phrase time-series.

4. Panels 2b, 2c, 3b, 3c. How is calculated the temporal correlation? between what and what?

We now specify in the text that the temporal correlations are performed between the time-series of the amplitude of the SOM patterns through time, and the spatial correlations are calculated across the SOM patterns themselves.

5. Line 207, "AQA is strongly correlated with Qf Pattern 2". AQA is a time series, but the Qf SOM pattern is a geographic structure. How can they be correlated?

To address the reviewer's concern, we explicitly stated that the correlation is between AQA and the temporal amplitude of the second Qf SOM pattern.

**Reviewer 2**

This paper focuses on an important challenge of improving seasonal prediction of winter precipitation in the Levant region by connecting it with Mediterranean Sea heat uptake patterns. The authors use Self-Organizing Maps (SOM) to identify three main spatiotemporal patterns and develop the Aegean Qf Anomaly (AQA) index representing SOM 2, which correlates strongly with Levant winter precipitation. The research provides valuable insights for seasonal prediction. I also liked that they backed their research with physical interpretation through hydrological decomposition and synoptic analysis.

We thank the reviewer for the productive and insightful comments.

While the overall approach is solid, I have several concerns about methodological choices and explanations that need addressing. Here are my main points:

Major comments

1- SOM vs. EOF choice:

-The authors mention SOM's advantage of not requiring orthogonality, but don't fully explain why this matters for this specific analysis.

-If EOF produces similar patterns as, including at least one EOF figure in the Supplement would strengthen this justification.

To address the reviewers' comments on our decision to focus on SOM analysis rather than EOF, we added a section in the SM comparing the EOF analysis of Mediterranean SST and ocean heat uptake, which shows that the EOF-based results are very similar, albeit slightly less robust. Our conclusion that only three key modes dominate Mediterranean Sea variability, therefore, is independent of our choice between SOM and EOF analysis. We present the SOM patterns because they present somewhat higher correlations with our target region (the Levant). We hypothesize that the stronger correlations among SOM patterns are due to their not being constrained to be orthogonal (as in EOF), allowing the leading SOM patterns to capture more of the Mediterranean variability.

2- SOM parameter selection:

-The optimization approach - maximizing correlation with precipitation while minimizing pattern count - seems somewhat methodologically questionable. Why is maximizing correlation with the target variable appropriate for what should be a self organising clustering technique?

Note that the optimization was not particularly sensitive to the SOM parameters (shown in Figure S1), so that the number of patterns is essentially the key optimization parameter (as also indicated by the similar emergent patterns in the EOF analysis). In that regard, it makes sense to aim for patterns that capture Levant winter precipitation variability with the fewest degrees of freedom. We now explain this rationale in the revised text and also added more details about the SOM structure selection approach to the SM.

3- AQA index definition

-The fixed Aegean Sea box used for the AQA index appears to be visually selected based on SOM Patterns 2 and 3. While it is an effective choice, I recommend discussing the robustness of the AQA box definition.

To address the reviewer's comment, we added a note on the sensitivity of the AQA index definition in Section S5 of the SM. Specifically, we adjusted the definition of AQA by adding and subtracting 0.25 degrees north-south and east-west and calculating the correlations of the different regions with Levant winter precipitation. We find that a difference of ±0.25 degrees to the north or to the east in the definition of AQA has no significant effect on its correlations with Levant winter precipitation.

-The key correlation between August AQA and winter precipitation (R = –0.60) needs more context. Please clarify exactly how Levant precipitation is calculated (sum/average over the region) and include confidence intervals for the correlations in Fig. 5b.

We added a figure to the SM showing the correlation statistics between AQA and Levant winter precipitation. It shows that the correlations in August AQA values are statistically significant at the 95% level.

The Levant land winter precipitation we analyze is calculated as the monthly weighted mean precipitation anomaly from climatology over the Levant domain identified in Figure 1. To ensure the reliability of the precipitation data, we also use in situ observations from meteorological stations around Israel. For the rain gauge data, we calculate the mean of all the stations within the domain.

5- Physical mechanisms

-While the composite analysis showing Cyprus Low persistence and subtropical jet strengthening is convincing, I would be cautious claiming independence from remote drivers (NAO, ENSO, etc.). Lack of correlation doesn't necessarily mean lack of physical connection. A mechanistic explanation would be more persuasive.

We agree with this critical point and now seek to clarify it in the Discussion. Specifically, while we show in Figure S15 that NAO and ENSO are not statistically related to AQA, we cannot exclude their potential influence in a more complex way.

Minor comments

- In abstract be specific that R = –0.60 refers to the correlation between August AQA and DJF Levant precipitation.

To address the reviewer's comment, we adapted the abstract to explicitly state that the correlation is between August AQA and DJF precipitation.

- In methods when describing heat uptake (Qf), clearly state that positive values mean the ocean gains heat and negative values mean heat transfer to the atmosphere.

We thank the reviewer for pointing out this issue. To make the manuscript clearer we explicitly stated in the text that positive heat uptake values indicate heat transfer from the atmosphere to the ocean.

- Figures 2–3: Include the cumulative variance explained by the three SOM patterns.

To address this comment, we added the variance explained by the 3 SOM patterns to the captions of Figures 2 and 3. Additionally, we added a figure in the SM (Figure S1) showing the variance explained by the different SOM structures of Qf and explaining its role in selecting the final SOM structure.

- Figure 5: Add statistical significance indicators or confidence bounds to the correlation values.

To avoid saturating the main-text Figure with detail, we added to the caption of Figure 5 a statement pointing to SM Figure S17 that shows the significance parameters (P-values and 95% upper and lower bounds) for the correlation coefficients between AQA and Levant mean anomaly land precipitation.

- Vague phrases like "optimal results" should be clarified - do you mean statistical robustness, strongest correlation, or something else?

In the revised text, we tried to avoid such vague terms. For example, we now describe the selection process for the number of SOM patterns, previously referred to as "optimization," in more detail.

**Reviewer 3**

The researchers investigated the seasonal predictability of winter precipitation in the eastern Mediterranean (EM) with respect to the Mediterranean SST and heat uptake in the preceding summer/autumn. Based on both reanalysis data and observations, the authors use a SOM algorithm to classify SST and heat uptake fields into three groups. of them two show correlation to the following winter EM precipitation. Spatial patterns of lagged correlations between SST and Qf and subsequent winter precipitation in the EM are explored and compared between the two relevant groups, detecting peaks in August at the Aegean sea. The authors then define an Aegean sea anomaly index (AQA) which, when taken for August months, can act as a precursor with negative correlations to winter precipitation in the EM. Composites of positive and negative AQA-preceded winters are then further analyzed, thoroughly investigating the impact of the AQA on synoptic systems and on the decomposed hydrological balance.

The researchers conclude by proposing a cross-seasonal link between Aegean SSTs and Qf in August to winter precipitation in the EM. Through the eastward migration of the subtropical jet following negative AQA anomalies, cyclonic activity is reinforced in the EM, allowing for more persistent precipitation.

The study pursues a relevant objective in a novel approach, using reliable datasets and a range of sound methods. The manuscript is comprehensive and sound, well written and nicely structured, albeit slightly dispersed and over-informative at times.

I recommend accepting the paper after some minor concerns are answered and revisions made.

We thank the Reviewer for the useful comments and suggestions, which have been implemented in the revised text and Supplementary Materials (SM), as detailed below.

General comments:
1. There is a general feeling that the MS is constructed step by step based on its own statistical results. I recommend restructuring the MS in light of the results and the decisions taken by them, in a way that wouldn't overwhelm the reader. E.g., if the SOM shows similar results for both SST and Qf, and seeing as Qf proves to be a better predictor, why not avoid showing and discussing SST throughout? Clearly a lot of effort was put into this work, but not all must be shown to the reader.

Thanks for pointing this issue out. As mentioned by Reviewer 1, SST is a state variable that has been used in previous works and is a more intuitive choice as a potential predictor for most readers. This, in part, is the rationale for showing the relation to SST variations in Figure 1. Moreover, ocean heat uptake (Qf) and SST are related such that for a fixed mixed layer depth, the tendency of SST is proportional to Qf. The fact that the lagged correlations for Qf are lagged by an additional month compared to SST (as seen in the comparison between Figures 2 and 3) therefore shows physical consistency which bolsters our results. We estimate that providing results for both SST and Qf variations would facilitate a more intuitive interpretation of our results and make them more comparable to previous work. Thus, while we acknowledge that presenting both SST and Qf results may encumber the presentation, we tried to provide an overall balanced presentation that focuses on Qf while providing relevant context for SST where needed. In the revised text, we try to address the reviewer's concern, and provide more context for the presentation of both SST and Qf.

2. The SOM algorithm is underutilized here. The 1X3 network inherently looks for zonal variability only, leading to a rough separation that could have been easily obtained with less complex algorithms such as EOF or k-means, simplifying the interpretation of the results. Also, a significant test / standard deviations for the classified patterns is required to show regions of higher and lower confidence.

For the analysis shown in this manuscript, we used both the SOM and EOF methods, focusing on the SOM analysis for the final version and publication, because it yields higher correlations between the emergent patterns and Levant winter precipitation. To ensure the robustness of the results, we added to the EOF analysis of both SST and ocean heat uptake to the supplementary materials, showing that similar results can be achieved using EOF (incorporating more patterns and slightly weaker correlations). Additionally, to better explain the optimization method for the SOM analysis, we elaborate in the supplementary materials section on the SOM analysis and optimization, including a significance test for the SOM patterns and additional details on the optimization process. The optimization process for the final SOM structure consisted of running the SOM analysis algorithm for different SOM configurations (changing the number of columns and rows) and calculating the amount of variance explained by all the patterns, as well as the amount of variance explained by the leading SOM pattern. Additionally, we also calculated the topographical error of the SOM structure. The final optimization method used in our analysis

was maximizing the amount of variance explained by the SOM structure and the leading SOM pattern, as well as minimizing topographic error and the overall number of patterns and repetitions within patterns. This optimization analysis yields the structure of 3 SOM patterns that we use in our analysis.

3. I think conciseness should be sought after in the revised MS. Some information can be removed from the figures and following discussions, increasing the focus on the main points the authors wish to convey.

This comment resonates with comment 1, and we agree with the reviewer that we should aim for a more focused and concise presentation. We tried to implement this guideline throughout the revised text.

Minor comments:
Figure 1: why is it important to show the difference between summer and winter precipitation? This panel isn't addressed in the text and the point of it is unclear.

Since Eastern Mediterranean precipitation occurs primarily in winter, winter minus summer highlights the Mediterranean regions, particularly the Levant. This is analogous to analyses of monsoonal regions, where summer minus winter precipitation is used to highlight monsoonal precipitation better. We clarify this in the revised text.

Also, the global maps are hard to see. Is it essential to have three of them? Perhaps one is enough to provide the context of remote teleconnections?

Figure 1 is used to facilitate discussion of previous similar works, examining the relationship between Levant winter precipitation and various climate indices worldwide. But the analysis of the specific influence of the Mediterranean remained lacking. We show the lagged responses to September, October, and November to demonstrate that, regardless of the specific details of the correlation for each month, the lagged relations vary significantly across months, pointing to complex dynamic influences (e.g., the tropical Atlantic and the Indian Ocean regions). In the revised text we try to better explain the motivation for Figure 1, while keeping the text as concise as possible.

L52: amplification of…

Corrected.

L179 and elsewhere: the authors refer to cluster frequency as "explained variance". I find this terminology more suitable for EOF analysis. For clarity and fluidity, consider using cluster frequency (or a similar term) throughout.

We calculated the amount of variance explained by incorporating both the pattern frequency, and the amount of variance explained by each pattern. The resulting variance explained is thus impacted by the pattern frequency, but is not identical to it. We now try to clarify this in the text and figure captions.

L184: what do you mean by "node"? please clarify

To address the comment, we removed the word "node" from the text and explained our meaning of the SOM structure and method in a more straightforward way.

Figure 2: I find the "correlation across SOM parameters" puzzling and unnecessary to propagate the reader through MS. Similarly, showing multiple months with mostly weak correlations that fluctuate from month to month does not support the robustness of the analysis. I suggest keeping the only the leading SOM input - Qf and only the most successful lead-time (August). This will make the message easier to take in.

We acknowledge that this may burden the reader and even obfuscate the results. We nevertheless elect to show the full set of results for completeness and transparency and try to do a better job of pointing the reader to the critical information.

L200: This is a methodological leap, especially since it is unclear how robust these spatial patterns are in the Aegean. More statistical testing is required to establish the Aegean Sea as an anchor for the SOM separation. E.G., did the results differ significantly when using the Ionian/Ligurian seas instead?

The Aegean Sea was selected because, consistent with the SOM patterns, it proved to be the strongest predictor of Levant winter precipitation. Thus, it was not selected based on SOM separation, but rather on its relation to Levant precipitation. We have extensively examined different regions that indicated potential predictive power (the vicinity of the Gulf of Lion in particular), but found the Aegean Sea to stand out in its statistical relation to Levant winter precipitation. We now state this in the text.

L207: anti-correlated?
To clarify that the correlation values are negative but significant, we changed the text to state that they are anti-correlated, as suggested by the comment.

Figure 4: would it be possible to denote the corresponding winter precipitation anomalies?

We have examined this. However, given that the winter precipitation anomalies consist of three-month averages and are related to AQA in a lagged manner, adding these on top of the AQAvariations  is not visually particularly revealing. We therefore elect not to display these in this figure.

L275: please state that these conclusions relate to the negative AQA phase (correct?)
Yes, the conditions described are relevant for winters preceded by negative AQA values in August. We changed the text accordingly to clarify our meaning.

L280-283: this sentence is unclear. Please rephrase
To better clarify our meaning, we rephrased the sentence, expressing that the moisture balance analysis's results showing the dynamical mean component as important align with the intensified jet we observe.

L283: what do you mean by "geostrophic enhancement"?
We now explain that this refers to increasing horizontal pressure gradients with height, caused by the thermal low.

L293: I would consider rewriting this section with emphasis on the importance of this work and less speculations and general ideas that do not relate directly to this research.

We tried to make the Discussion less speculative and more focused on the results.

[revised manuscript text omitted]

---

## Referee Report (RR1)

Mediterranean Sea heat uptake variability as a precursor to winter precipitation in the Levant – review

The authors elaborate on the SOM configuration and EOF analysis. However, this analysis raises several additional issues:

1. The approach of the authors towards the SOM analysis gives the impression that it is being mistreated: SOMs do not have a "leading pattern" and are not constructed to minimize the explained variance, nor do they have temporal "amplitudes". Rather, SOM nodes represent cluster centroids and each sample (in this case, monthly anomaly) is assigned to a single pattern, unlike EOF where each sample is constructed by different amplitudes of the EOF modes. SOM nodes are not directional vectors that explain temporal variations either, unlike EOFs. The authors seem to be referring to node frequency as amplitude, in which case it is unclear over which period this frequency is calculated.

2. The supplementary material does not explain the choice of 1X3 SOM configuration using a relevant measure: this should include an elbow-method analysis or minimizing the SOM quantification error. If anything, results shown in the supplementary should motivate the authors to enhance the SOM to at least a 2X3 configuration, drastically raising the total explained variance (though this is not a traditional requirement from a SOM analysis) and obtaining refined spatial patterns. E.g., the key area of the Aegean sits in the low-confidence area of node 3 – suggesting that the association between the AQA and the SOM nodes is weaker than it seems from the composite maps. This may improve with a refined SOM application.

3. Maximizing the variance explained by the "leading" SOM pattern counteracts the primary purpose of the SOM – building clusters with minimal internal variability.

4. Topographic errors in 1D and 2D SOMs are incomparable – the topographic error is asking how many of 2$^{nd}$-winning neurons are not neighbors of the 1$^{st}$ winning neurons. E.g., in the selected 1X3 SOM most neurons are neighbors by construction, and so the TE of this configuration is not comparable to the 2D configurations and does justify the choice.
TEs are used to measure the continuity of the SOM space – e.g., are there non-neighboring neurons that are very similar to each other. For selecting SOM size, the quantification error is more relevant, and even that is only borderline comparable between 1D and 2D SOMs, as 1D networks inherently emphasize one dimension of variability – more suitable for analyzing temporal variability of a local time-series, etc.

5. Moreover, if the conclusion is that EOFs can readily produce similar correlations and seeing that the SOM algorithm is underutilized and treated as an EOF analysis throughout, I recommend switching to the EOF results for clarity.

   In my view, SOM is meant to enter where EOF falls short of capturing the dominant patterns of the system, or if higher precision is sought after. However, this SOM configuration essentially converge to the EOF results – rendering its inclusion redundant.

   Changing to EOF analysis will highlight the importance of the dynamical pathway described here rather than focusing on the clustering approach – which involve several subjective choices that are not justified well by the authors. I believe that EOF serves a more objective, reproduceable, and physically interpretable approach for the purpose of this MS.

Minor comments:

1. The global maps in figure 1 are redundant for the purpose of this MS. If a case is to be made concerning them, it can surely be made using a single map.
2. There is no justification to display and discuss two SOM networks with highly similar results (e.g., SST and Qf). Choose one, and state that similar results are obtained if the other is used. This is not very surprising seeing as the two fields are highly correlated.
3. Most references do not include a doi, making the review process unnecessarily tedious, and are not in line with the WCD format requisites.

I recommend accepting the MS once the issues with the SOM analysis are resolved – either changing entirely to EOF framework or enhancing the SOM analysis to justify its use.

---

## Author Response (AR2)

**Mediterranean Sea heat uptake variability as a precursor to winter precipitation in the Levant**
Cohen et al.

**Response to Reviewer 3:**

We thank the reviewer for his helpful comments and his insistence on simplifying and improving the presentation of the results. Specifically, we accept the Reviewer's suggestion of change the focus in the manuscript to the EOF analysis and results, instead of the previous focus on the SOM analysis. We therefore moved the EOF-based and results from the Supplementary Materials to the main text, and moved the SOM-based results to the Supplementary Materials instead. This change has no implications for the main results and conclusions, while significantly improving the clarity of the presented results. Furthermore, to address the additional comments raised by the reviewer about the SOM analysis and its parameter selection, we amended the explanation about the SOM analysis and results in the supplementary materials. Specifically, we discuss the limitations of SOM analysis and the fact that it is not designed to maximize the amount of variance explained, and to emphasize that we tried to find the minimal number of patterns beyond which repetition in the patterns' characteristics emerges.

A comment-by-comment (blue) response (black) is provided below:
The authors elaborate on the SOM configuration and EOF analysis. However, this analysis raises several additional issues:
1. The approach of the authors towards the SOM analysis gives the impression that it is being mistreated: SOMs do not have a "leading pattern" and are not constructed to minimize the explained variance, nor do they have temporal "amplitudes". Rather, SOM nodes represent cluster centroids and each sample (in this case, monthly anomaly) is assigned to a single pattern, unlike EOF where each sample is constructed by different amplitudes of the EOF modes. SOM nodes are not directional vectors that explain temporal variations either, unlike EOFs. The authors seem to be referring to node frequency as amplitude, in which case it is unclear over which period this frequency is calculated.
As the Reviewer suggested, we now base our results on the EOF analysis, and use the SOM analysis as support for the robustness of the results. There is therefore no need to consider the emergent SOM patterns as 'leading patterns'. Instead, the emergent three

SOM patterns nearly match the first 3 EOF patterns, indicating that our results are not sensitive to the choice of methodology. We explain that SOM pattern loading can be regarded as analogous to the principal component of the EOF, as evidenced by the similar correlation results for the EOF and SOM patterns.

2. The supplementary material does not explain the choice of 1X3 SOM configuration using a relevant measure: this should include an elbow-method analysis or minimizing the SOM quantification error. If anything, results shown in the supplementary should motivate the authors to enhance the SOM to at least a 2X3 configuration, drastically raising the total explained variance (though this is not a traditional requirement from a SOM analysis) and obtaining refined spatial patterns. E.g., the key area of the Aegean sits in the low-confidence area of node 3 – suggesting that the association between the AQA and the SOM nodes is weaker than it seems from the composite maps. This may improve with a refined SOM application.

Given that the focus has shifted to the three leading EOF patterns, there is no need to justify the choice of 3X1 SOM structure. We nevertheless mention in the Supplementary Materials that pattern redundancy appears for structures larger than 3X1.

3. Maximizing the variance explained by the "leading" SOM pattern counteracts the primary purpose of the SOM – building clusters with minimal internal variability.

We now avoid using this terminology.

4. Topographic errors in 1D and 2D SOMs are incomparable – the topographic error is asking how many of 2nd-winning neurons are not neighbors of the 1st winning neurons. E.g., in the selected 1X3 SOM most neurons are neighbors by construction, and so the TE of this configuration is not comparable to the 2D configurations and does justify the choice. TEs are used to measure the continuity of the SOM space – e.g., are there nonneighboring neurons that are very similar to each other. For selecting SOM size, the quantification error is more relevant, and even that is only borderline comparable between 1D and 2D SOMs, as 1D networks inherently emphasize one dimension of variability – more suitable for analyzing temporal variability of a local timeseries, etc.

Given the focus on EOF patterns, the discussion of topographic error in the SOM patterns is no longer relevant.

5. Moreover, if the conclusion is that EOFs can readily produce similar correlations and seeing that the SOM algorithm is underutilized and treated as an EOF analysis throughout, I recommend switching to the EOF results for clarity. In my view, SOM is meant to enter

where EOF falls short of capturing the dominant patterns of the system, or if higher precision is sought after. However, this SOM configuration essentially converge to the EOF results – rendering its inclusion redundant.

Changing to EOF analysis will highlight the importance of the dynamical pathway described here rather than focusing on the clustering approach – which involve several subjective choices that are not justified well by the authors. I believe that EOF serves a more objective, reproduceable, and physically interpretable approach for the purpose of this MS.

We thank the Reviewer for insisting on this and have accepted the suggestion.

Minor comments:

1. The global maps in figure 1 are redundant for the purpose of this MS. If a case is to be made concerning them, it can surely be made using a single map.

We accept the Reviewer's suggestion. We now present only one global correlation map in Figure 1.

2. There is no justification to display and discuss two SOM networks with highly similar results (e.g., SST and Qf). Choose one, and state that similar results are obtained if the other is used. This is not very surprising seeing as the two fields are highly correlated.

Given that the tendency of SST is related to $Q_f$, the relation of these two fields, as well as their relation to Levant precipitation, is nuanced. We elect to show both fields because most previous work has focused on SST, whereas our results indicate that $Q_f$ is likely a more relevant field. To address the Reviewer's consent, we have now joined the presentation of the SST and $Q_f$ results into a single figure, which allowed clarifying the text discussing the SST and $Q_f$ patterns in Section 3.1.

3. Most references do not include a doi, making the review process unnecessarily tedious, and are not in line with the WCD format requisites.

DOI added to references.

I recommend accepting the MS once the issues with the SOM analysis are resolved – either changing entirely to EOF framework or enhancing the SOM analysis to justify its use.

As suggested, we now base our analysis on the EOF framework.